# Inverse Kinematics: An Alternative Solution Approach Applying Metaheuristics

**Raúl López-Muñoz** [1], **Edgar A. Portilla-Flores** [2], **Leonel G. Corona-Ramírez** [3], **Eduardo Vega-Alvarado** [1,*] and **Mario C. Maya-Rodríguez** [4]

1   Group of Research and Innovation in Mechatronics (GRIM), Centro de Innovación y Desarrollo Tecnológico en Cómputo (CIDETEC), Instituto Politécnico Nacional, Mexico City 07700, Mexico
2   Group of Research and Innovation in Mechatronics (GRIM), Unidad Profesional Interdisciplinaria de Ingeniería Campus Tlaxcala (UPIIT), Instituto Politécnico Nacional, Tlaxcala 09000, Mexico
3   Unidad Profesional Interdisciplinaria en Ingeniería y Tecnologías Avanzadas (UPIITA), Instituto Politécnico Nacional, Mexico City 07340, Mexico
4   Escuela Superior de Ingeniería Mecánica y Eléctrica (ESIME), Instituto Politécnico Nacional, Mexico City 07738, Mexico
*   Correspondence: evega@ipn.mx

**Abstract:** The inverse kinematics problem (IKP) is fundamental in robotics, but it gets harder to solve as the complexity of the mechanisms increases. For that reason, several approaches have been applied to solve it, including metaheuristic algorithms. This work presents a proposal for solving the IKP of a doubly articulated kinematic chain by means of a modified differential evolution (DE) algorithm. The novelty of the proposal is both in the modeling of the problem and the modification to the DE for solving it. The modeling is inspired by a technique used in animation software to recreate movements by dividing the complete trajectory in a number of segments. Each segment represents a single optimization problem linked to the IKP as a sequence that is solved by the modified DE where the initial population for each single problem is biased by using the solution of the previous one. The approach produces solutions for positioning the end effector in a specific point within the work space while minimizing the angular displacement between the initial and final poses. The proposal was able to obtain solutions requiring a fewer total execution cycles compared to the usual approach of solving only one optimization problem related to the inverse kinematics. Different trajectories were used to test the solutions generated by the proposed approach, and the set of conditions that must be covered to apply it to solve the IKP of a particular mechanism are presented.

**Keywords:** inverse kinematics problem; optimization; metaheuristic algorithm; articulated kinematic chain

## 1. Introduction

The specific positioning of a kinematic chain is a recurrent problem in different areas of engineering such as robotics, animation, or in articulated reference models for graphic illustrators. In many cases, it is possible to achieve the desired position by moving each link through angular movements until the proper posture is achieved. However, in other occasions, a method is desirable or even necessary that can position the rest of the links to meet a specific condition, such as positioning the end effector of the last link, which is known as inverse kinematics (IK).

The process of finding the IK is complex [1–3], and for this reason, diverse ways to deal with this task have been explored: analytic methods, neural networks, and metaheuristic algorithms. Traditional approximations, as is the case of obtaining the analytical equations via geometric analysis, are limited to working with kinematic chains of 2-, 3-, or, in some cases, 4-DOF at maximum, and their solution requires the use of constraints. This is because one of the aspects that increases the complexity of finding the IK is the number of degrees of freedom, since there are multiple solutions that satisfy the desired conditions.

When it is required to solve hard optimization problems, especially on real-world engineering cases, metaheuristic algorithms are a good option when traditional methods present deficient performance. Metaheuristic algorithms showed good performance when solving the IKP without being affected by the number of degrees of freedom, as shown in [4]. In general, metaheuristic algorithms are iterated approximate numerical techniques that involve different processes with stochastic variables. In these methods, it is necessary to adequately model the problem with a fitness function to quantify the quality of the candidate solutions. Since they are approximate techniques, the optimal solution is not necessarily found, understanding this as the minimal or maximal value of the function depending on the context, but good-enough solutions can be obtained. For example, in [4], in addition to proposing a hybrid method between two algorithms, namely, DE [5] and particle swarm optimization (PSO) [6]), the authors presented a series of proposals to achieve the desired position and additional objectives: minimal displacement between solutions, collision avoidance, and functional joint relations.

As in [4,7], most of the works based in metaheuristic algorithms use the obtained equations from homogeneous transformations following the Denavit–Hartenberg convention [8] as a means to formulate the performance function in order to minimize the error between the desired and real positions mapped by the forward kinematics (FK) model. In [2], a modified genetic algorithm was proposed to obtain the IK of a 6-DOF robot, highlighting the modification performed to balance exploration and exploitation, which refers to the behavior of the algorithm to explore solutions throughout the search space, and to improve the candidate solutions. With this same idea of improving the performance of search algorithms, so-called hybrid algorithms were produced that take steps or operators from other algorithms and adapt them to modify their own behavior. In [9], the authors presented the hybrid mutation fruit fly optimization algorithm in which they implemented an olfactory search on the basis of different mutation strategies from different DE versions. The performance of the proposed algorithm was evaluated using a problem to obtain the IK of a manipulator with 7-DOF and 8 benchmark functions.

In [10], the IKP was solved with a modified PSO to increase the speed of convergence through a bidirectional search and the decoupling of the kinematic chain while considering the characteristics of the robot, a 4-DOF manipulator with a mobile base. In other cases, in addition to obtaining a solution close to the optimum, it is necessary to generate it in a given time, which, in turn, implies the fast convergence of the algorithm. For this reason, the authors in [11] proposed a modification to DE in order to reduce the required time for finding the IK of a robot with 3-DOF, introducing a local search mechanism that they called discarding. Memetic algorithms [12–14], are hybrid methods that take advantage of the synergy from the combination of the global search capacity of some technique and the power of the local search of another mechanism. This paradigm was used in [4] for solving the IKP of different robots with 6- and 7-DOF as a means of accelerating the convergence.

In the aforementioned works, emphasis was placed on modifying the metaheuristic algorithms to solve the IKP, obtaining good results while increasing the convergence speed. In this proposal, the modifications to increase the calculation speed are applied to modeling the optimization problem and the generation of the sets of proposed solutions. For the first point, the total trajectory to be tracked by the robot is divided into a series of points, with each pair of consecutive points representing a small trajectory by itself. The IKP solution for each point is solved as an individual optimization problem, reducing the complexity of the problem and accelerating the solution. For the second point, the last generated solution is good enough to be taken as the base to generate the new population for the next optimization problem, also accelerating the solution. So, the initial population for each problem is biased by using the solution of the previous one, thus modifying DE and its behavior.

In order to evaluate the performance of the IKP solution method presented in this development, we compared well-known approach presented in the related literature that consisted of directly finding the IK from a given performance function, and the proposed

approach in which the original problem is divided into a set of simpler problems, in addition to manipulating the initial population in each of them. As an additional feature, every solution has a minimal displacement compared to its original pose. Achieving this characteristic is especially useful in animation or simulation fields where there is a special interest in visualizing how a particular movement would be regardless of its causes, since each solution represents an instant of the possible movement to be carried out to produce the desired positioning. In the field of animation, the cyclic coordinate descent algorithm (CCD) [15] is used in different dedicated software and in video game engines such as Unity because it can find good solutions in a short time. Despite this, the authors in [16] reported that it also presents problems such as generating inadequate rotations under certain circumstances, a problem that is avoided in this proposal because of the aforementioned generation of the new populations.

This work is organized as follows: Section 2 presents the proposed solution for the IKP and the corresponding case study. Section 3 describes the conducted experiments and their results. Lastly, Section 4 consists of the obtained conclusions from the experimental data and possible future works.

## 2. Materials and Methods

This development is aimed at solving the IKP of kinematic chains with five or more DOF, since analytical methods present complications as the degrees of freedom increase. In the case of the cited works that use metaheuristics, the kinematic chains of 5-, 6-, and 7-DOF refer to industrial robots that can be found in the real world. In this work, the requirements of other areas such as animation are considered, but the proposal can also be applied to solve the IKP of industrial robots.

In areas related to computer graphics, kinematic chains do not refer to models that deal with physical constraints. For this reason, they can be more complex in the sense of the number of DOF. Solving their IKP is still a necessity, ideally in a short time and, in most of the cases, with the additional characteristic of minimizing the angular displacement of the effectors between the original and last positions. For this reason, the case study for this proposal is an open kinematic chain with five links that were linked with joints that could rotate with respect to two axes, each joint with 2-DOF. This implies that, for a position on the XYZ coordinate system, it is required to find 10 angular values.

The FK of the case study was modeled taking into account the Denavit–Hartenberg convention [8], such that the desired coordinates were the first three elements of the last column in the matrix in (1) ($\vec{P}$), resulting from the multiplication of the homogeneous transformations shown in (2) and considering the parameters of Table 1.

$$H = \begin{pmatrix} R & \vec{P} \\ 0 & 1 \end{pmatrix} = H_1 H_2 H_3 \ldots H_{10}, \tag{1}$$

$$H_i = \begin{pmatrix} \cos\theta_i & -\sin\theta_i\cos\alpha_i & \sin\theta_i\sin\alpha_i & l_i\cos\theta_i \\ \sin\theta_i & \cos\theta_i\cos\alpha_i & -\cos\theta_i\sin\alpha_i & l_i\sin\theta_i \\ 0 & \sin\theta_i & \cos\alpha_i & d_i \\ 0 & 0 & 0 & 1 \end{pmatrix}, \tag{2}$$

**Table 1.** Parameters for homogeneous transformations.

| Links | $l_i$ | $\alpha_i$ | $d_i$ | $\theta_i$ |
|---|---|---|---|---|
| 1, 3, 5, 7, 9 | 0 | $\frac{\pi}{2}$ | 0 | $q_1$, $q_3$, $q_5$, $q_7$, $q_9$ |
| 2, 4, 6, 8, 10 | 10 | $-\frac{\pi}{2}$ | 0 | $q_2$, $q_4$, $q_6$, $q_8$, $q_{10}$ |

### 2.1. Optimization Approach

Using as a base the description of the FK and the space of configurations of the kinematic chains (the range of values that the intermediate effectors can take), it was

possible to implement an optimization function that quantified as an error the Euclidean distance between a mapped solution corresponding to the obtained FK model and the desired position.

The general description of a numerical optimization problem is expressed in (3), where $f(\vec{q})$ is the function to minimize that depends on the vector of design variables $\vec{q}$. This function can be subject to inequality and equality constraints $g_i(\vec{q})$ and $h_j(\vec{q})$, respectively.

$$\min f(\vec{q}) \tag{3}$$
$$g_i(\vec{q}) \leq 0$$
$$h_j(\vec{q}) = 0$$

For this case study, the objective function is described in (4), where $x_d, y_d, z_d$ are the desired coordinates, and $x(\vec{\theta}), y(\vec{\theta}), z(\vec{\theta})$ are the mapped coordinates from the FK corresponding to the $\vec{P}$ column of the resulting matrix in (1). The function was calculated using the design vector $\vec{\theta}$ that contained the 10 joint angular values. The value of the objective function is expressed in length units that depend on the particular case study.

$$\min f(\vec{\theta}) = \sqrt{(x(\vec{\theta}) - x_d)^2 + (y(\vec{\theta}) - y_d)^2 + (z(\vec{\theta}) - z_d)^2} \tag{4}$$

This optimization problem does not consider any constraint because the development was focused on the solution of the IKP problem for a kinematic chain without taking into account the usual physical events that are modeled as constraints such as collisions, and the search space is infinite in the sense that the joints have free rotation. In addition to positioning the end effector, it is desirable that the final posture adopted by the robot when reaching the desire position has minimal displacement with respect to the original pose. This is quantified by the Euclidean norm of the difference between the vector of angular values at the initial position ($\vec{\theta}_i$) and the vector $\vec{\theta^*}$ obtained by the proposed solution algorithm. In other works like [4], this norm appears explicitly in the optimization problem, but with the modification proposed in this work, it is not necessary to achieve solutions with a similar initial posture.

### 2.2. Selection of the Metaheuristic Technique

Diverse proposals have been developed, applying metaheuristic algorithms to solve the IKP. The proposal in this work is not to improve those algorithms, but to rethink the modeling of the problem and take advantage of its special characteristics to simplify the search work carried out by the metaheuristic algorithm. However, it is still necessary to select a metaheuristic as a starting point for the solution. In this sense, three widely cited algorithms in the related literature were considered that obtained good solutions in real-world engineering problems. These algorithms are DE, PSO, and artificial bee colony (ABC).

In this development, the tuning parameters, involved operators, and general performance were considered to select the algorithm to be used. With respect to the first criterion, only three parameters need to be tuned in DE, with well-defined intervals, and there is also available information on the impact of different values in the search. The number of individuals in the population is a parameter that shares with the rest of the algorithms. PSO has five parameters: only two are bounded in a well-established interval, and another is associated with the inertia of the particle, which can be interpreted as a particle resistance (candidate solution) to change its search direction, but an inappropriate selection can cause the population to diverge. In addition to this, unlike the rest of the algorithms that only require initializing the individuals, it is also necessary to initialize a set of process speeds that is still discussed as carry out to achieve good results. ABC has only two parameters for determining when a solution is randomly reset, which is an implemented mechanism due to the designer being inspired by the behavior of the bees when a food source (solution) runs out.

In the case of ABC, the random restarting of a solution after its limit of improvement trials has been reached is a mechanism that improves the global search capacity, increasing the probability to escape from local optima. Although it is an advantage in most problems, it is an obstacle to one of the main adaptations proposed in this paper. It consists of using the information from previous solutions to guide the next population by transforming the optimization problem into a sequence of simpler problems whose consecutive solutions are expected to be similar. So, it is desirable to bound the set of solutions. The opposite is the case of DE, which converges quickly and does not adequately explore the entire search space, but as mentioned, this has little impact because it is assumed that the solution to a problem of the sequence of optimization problems is already a good solution for the next one, and it requires a smaller space to explore. In the PSO algorithm, there is no particular behavior because it can be manipulated with tuning, but the base algorithm is usually modified to achieve good results.

Lastly, in [17] DE, PSO and ABC solving 24 benchmark problems [18] were compared, and one engineering design problem was presented, with DE occupying the first place in performance. Due to the compatibility of the proposal of using biased populations, having few tuning parameters and its performance in general, it was decided to use DE for the optimization stage.

### 2.3. Differential Evolution Algorithm

The DE algorithm is an optimization technique within the classification of evolutionary algorithms. It is a stochastic search method developed in 1995 by Kenneth V. Price and Rainer M. Storn [5], and several versions of DE have been developed. In this proposal, the variant rand/1/bin was used to obtain the solution of the optimization problem associated to the IKP. DE requires a population of individuals, each representing a proposed solution, and it includes process of mutation, crossing, and selection to evolve and improve it.

In evolutionary algorithms, mutation is a change in a pseudorandom element in order to maintain the diversity in the population. For the implemented DE version, a mutant vector ($V^j$) is created for each individual in the population following (5), using a target individual ($x^{r1}$), two additional individuals ($x^{r2}, x^{r3}$), and a mutation factor $F$.

$$MutantV^j = x^{r1} + F(x^{r2} - x^{r3});\tag{5}$$

The recombination is an operation in which test vectors are generated from the pseudorandom copy of the genetic material of the parent or mutant vector. The binomial cross process consists of selecting a random number between 1 and the dimension of the solution vectors; in this index, the child vector inherits the characteristics of the mutant vector. For the rest of the indices, the selection of the value to inherit is random and conditioned to (6), where $CR$ is the crossover.

$$NewV_i = \begin{cases} MutantV_i, & rand(0,1) \leq CR \\ FatherV_i, & rand(0,1) > CR \end{cases}\tag{6}$$

Lastly, a selection to update the population is performed following (7), where $f$ is the optimization function (4):

$$V^{g+1} = \begin{cases} NewV_i, & f(NewV_i) \leq f(V^g) \\ V^g, & other \end{cases}\tag{7}$$

### 2.4. Modeling Considerations

As mentioned before, the traditional approach of solving the IKP by finding a solution that minimizes the proposed error function (4) with an optimization algorithm was modified in this work, taking some ideas from the function of illustration software such as CLIP STUDIO PAINT (CSP) [19], where it is possible to manipulate articulated reference models, and there is the need to have different instances of the movement to reach the final posture.

The first modification consists of transforming the optimization problem into a sequence of problems, such that the solution of each of them represents an instant of the possible trajectory to reach the desired position. The trajectory can be proposed by the user or determined by an algorithm. This modification is an approximation to the way that CSP functions. When this software positions the end of a kinematic chain, the program iterates along with the user's displacement, such that it does not find a trajectory to the final destination, but solves according to the points where the mouse is moved, eventually reaching the desired position.

The second change was performed considering the implications of the first, the specific problem, and the way the differential evolution algorithm works. This modification consists in biasing the initialization of individuals: instead of initializing them randomly in the entire search space, they are initialized around a candidate individual (*pbest*) that is the solution of the previous problem, as described in (8), where $M$ is a user-defined factor that represents a search distance.

$$pop_k = pbest_{k-1} + M(rand - 0.5) \tag{8}$$

This can be achieved because it is assumed that the obtained solution for the previous problem after discretizing the trajectory would have an error that is at most equal to the resolution of the discretization. In addition, the solutions generated around that particular individual are similar to that. That is, the Euclidean norm of their difference is small, this being an additional desired aspect to the solution of the optimization problem.

### 2.5. Methodology Description

Figure 1 shows the proposed methodology. The first step is the formulation of the IKP as an optimization problem. Then, a trajectory is established between the first configuration (origin) and the final position. The trajectory is discretized with a sequence of points, and the segment between two consecutive points corresponds to a single optimization problem.

The third step consists in initializing the parameters for each problem, desire position, and candidate solutions. The population is biased around the solution obtained for the previous problem, with the exception of the first problem, where it is biased considering the origin.

The next steps are related to the optimization technique—in this case, DE. The fourth step consists in evaluating the current population (set of solutions) in each cycle. If the end criteria are met, the process for this segment finishes, and the algorithm continues with the next optimization problem. In the case of the end criteria not being met, in the fifth step, a new solution is generated for each solution in the population, applying (5) and (6). Lastly, at the sixth step, the individuals are compared using (4), and the best is kept for the next cycle, while the other one is discarded.

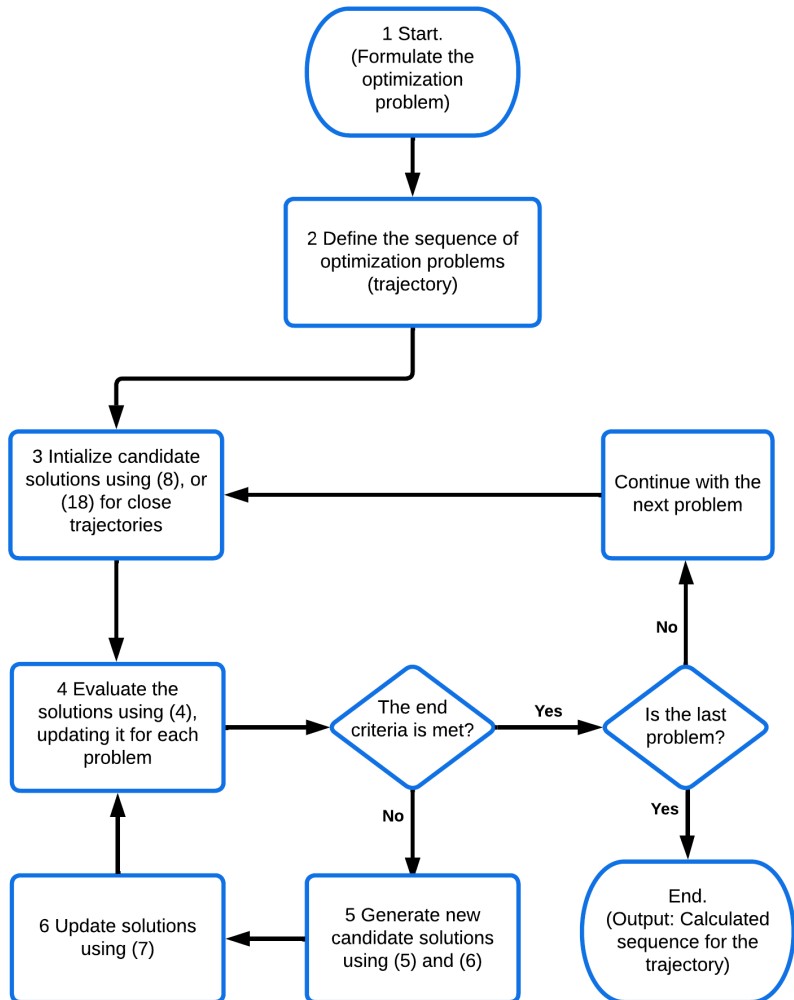

**Figure 1.** Flowchart for solving the IKP as a sequence of optimization problems.

## 3. Experiments and Results

In this section, the performed experiments are described, and their results are analyzed. All of them had in common a general case study that consisted in the solution of the IKP for a kinematic chain of five links, each of them 10 cm long and with double articular joints. This implies that the full kinematic chain had 10 DOF, and each joint is represented in Figure 2, considering that the distance between the $X_0, Y_0$ plane and $q_1$ was zero, so the kinematic chain was equivalent to coupling five of those robots sequentially.

The proposed method to solve the IKP is flexible and has the potential to be applied to different types of kinematic chains, and not only the general case proposed, as long as the conditions expressed in (9)–(14) are fulfilled. However, the method may not be applicable in the case of parallel robots, since in some of their configurations, there is no function to obtain the position and orientation coordinates of the end effector from the values angles of the actuators (FK), so the set of equations is not fulfilled.

$$S = \{(\theta_1, \theta_2, \ldots, \theta_n) | |\theta_i| \leq \theta\}; \tag{9}$$

$$\forall s \in S \exists f(s) | f(s) \in \mathbb{R}^3; \tag{10}$$

$$C = \{c = (c_1, c_2, \ldots, c_n) | c_i \in \{-e, 0, e\} | e > 0 \wedge ||c|| = e\} \tag{11}$$

$$\forall s \in S \exists c \in C | s + c \in S; \tag{12}$$

$$PS = ((x_1, y_1, z_1), (x_2, y_2, z_2), \ldots, (x_n, y_n, z_n)), \tag{13}$$

$$\forall (x_i, y_i, z_i) \in PS \exists s \in S \wedge |f(s_1) - ps_k| \leq \delta; \tag{14}$$

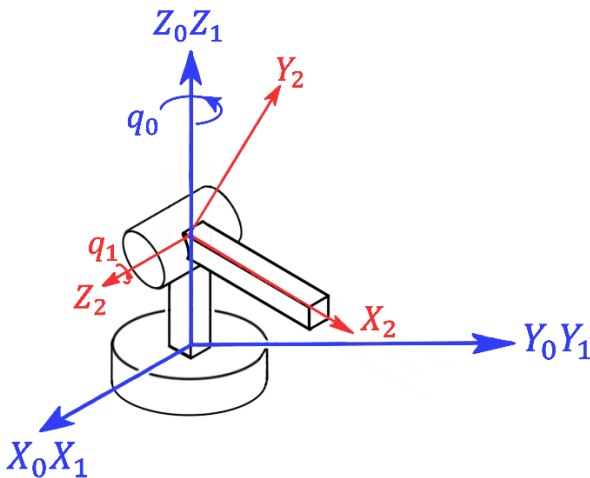

**Figure 2.** Robotic link with 2 DOF.

As can be seen, the configuration space $S$ must be described using a function that maps to $\mathbb{R}^3$, where $\theta_i$ refers to the value that the effector $i$ of the kinematic chain can adopt. The possibility of an increment or decrement of that value also needs to exist in a way that the new posture is part of the same space. In other words, there exist a set of changes ($C$) allowed for each DOF that must lead to a configuration in the same space, so for a path defined by a series of points ($x_i, y_i, z_i$), there exists a configuration that maps to a value in $\mathbb{R}^3$ with a boundary error (zero in the case that the point is in the work space and the effector adopts continuous values).

The FK models that were obtained by following the Denavit–Hartenberg algorithm fulfilled the (9)–(14). This is an advantage, but it also implies the limitations of the proposal, because it is applicable to the characteristics of IKP for robots or animation applications where an initial posture and the final value (not the solution) of the optimization function are met, and the final and initial postures are similar. These characteristics contrast with the benchmark problems shown in [18,20,21], where the optimal value was assumed to be unknown; in some cases, the best value was not necessarily the optimal, and it was not required that the final solution has similarity with a specific initial design vector.

Four experiments were carried out using this general case as a starting point. The first two are referred to as the common problem of finding the IK for a random point within the work space, and the difference between these experiments is the applied approach to find the solution. In the first experiment, the IKP was solved using DE as a mechanism to find the solution of a single optimization problem associated with the IK. In the second experiment, DE used biased populations to find the solution of a series of optimization problems, with each corresponding to an instant of the trajectory that the kinematic chain would follow to reach the desired final position.

After the comparison between these approaches and considering the results, the third and fourth experiments were carried out. They consisted of trajectory tracking applying the second proposal to determine its potential as a mechanism to obtain the configurations to be followed on a path that reproduces a consistent animation. This implies that the difference between two postures in a row must be minimal following the principles of the stop-motion animation [22]. In addition, the results of this experiment show that the second approximation presents robustness since it is not affected by the type of trajectory, maintaining a similarity between the initial and the final postures.

All experiments were performed using MATLAB R2020a software on a computer with the following specifications: Intel i7 processor @3.70 GHz, 16 GB RAM, and Windows 10 operating system. In addition, the tuning parameters for DE were: mutation factor $F = 0.6$, crossover probability $CR = 0.5$, and $pop = 10$ individuals throughout the generations.

### 3.1. Obtaining the IK for a Point in the Space

For the first two experiments, the objective point to position the final effector in the kinematic chain was selected as (20, −20, −10), in *XYZ* coordinates.

### 3.1.1. First Experiment

As previously explained, the first experiment consisted in solving the IKP as a single optimization problem. Ten executions of the algorithm were carried out taking as a stop criterion a limit of 100,000 generations or to obtain a solution whose error function is lower than $0.001 = 10 \times 10^{-3}$ cm. The obtained results are shown in Table 2, including the mean and the standard deviation of the results, where the best results are in bold type. In addition to the objective function, the similarity between the final and the initial configurations was calculated by using the norm of the difference of the initial solution and the final posture, where the similarity is inversely proportional to the distance. Since there was no an explicit mechanism that minimized the difference, the general similarity was low.

**Table 2.** Results of the first experiment.

| Run | Objective Function (cm) | Distance to the Original Pose (rad) |
|---|---|---|
| 1 | 0.0741 | 5.8277 |
| 2 | 0.0915 | 5.6552 |
| 3 | 0.1670 | 9.5290 |
| 4 | 0.0997 | 4.5117 |
| 5 | 0.0626 | 10.0037 |
| 6 | 0.0480 | 4.8227 |
| 7 | 0.0948 | 23.2495 |
| **8** | **0.1025** | **3.1664** |
| 9 | 0.1248 | 9.1732 |
| **10** | **0.0232** | **3.4065** |
| Mean | 0.0888 | 7.9345 |
| Standard deviation | 0.0403 | 5.6318 |

For this experiment, no solution was obtained in any of the 10 executions that met the error criterion before reaching the stop criteria. The result with the smallest distance from the initial pose was generated in the 8th run, whose angular values of the kinematic chain were 1.2055, −1.8763, −0.3180, 0.1339, −1.3818, 0.1600, −0.8093, −0.4621, 0.7072, −0.7136, while the run that best positioned the end effector was the 10th, whose values were 2.2508, −1.5886, −1.3051, −0.4985, −0.1481, −2.0907, 0.0696, 0.7274, 0.0150, 0.8789. The kinematic chain adopting the angular values of the solutions and the initial pose is shown in Figure 3. All the figures showing the results of the different experiments were generated using MATLAB R2020a.

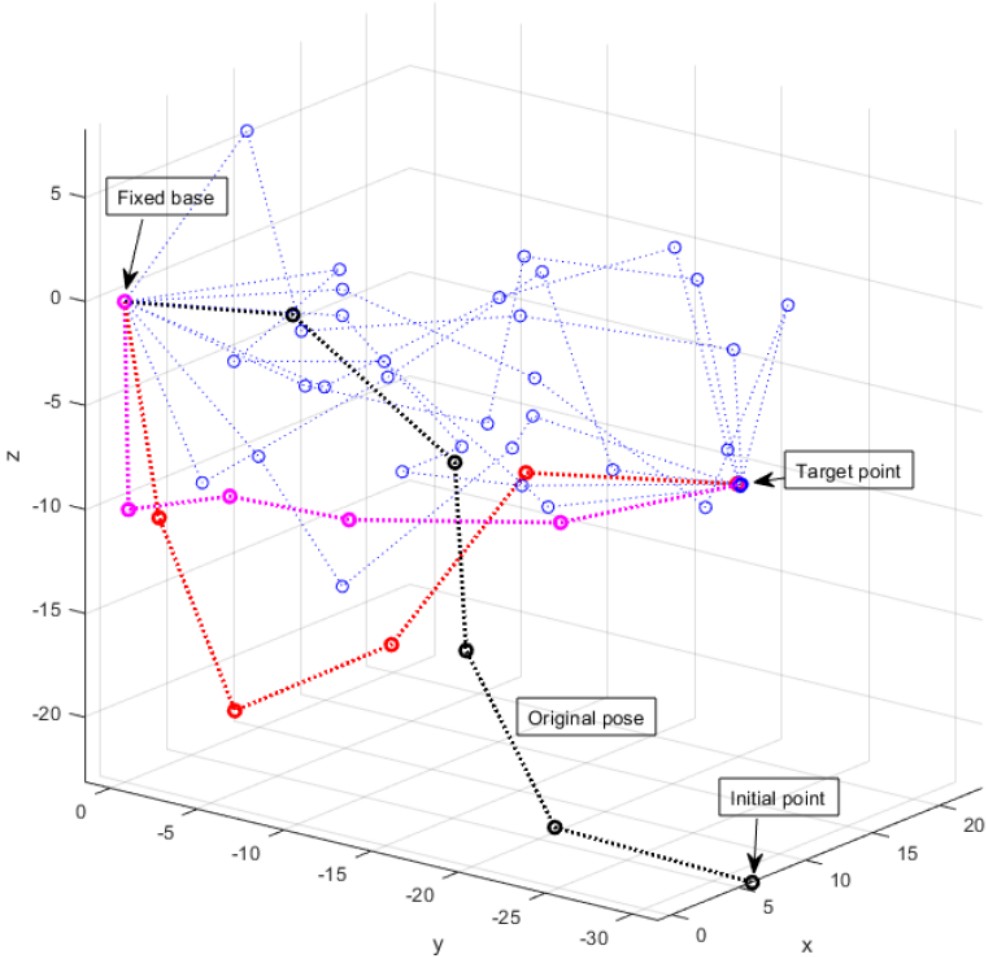

**Figure 3.** Kinematic chain when the solutions obtained in the first experiment were adopted. Original pose is shown in black, the pose with the greatest similarity in red, and the pose with the lowest error in magenta.

3.1.2. Second Experiment

In this experiment, the trajectory connecting the initial and the desired positions was discretized. So, the final solution was obtained on the basis of a series of $k$ optimization problems, with each associated with the IK of a small segment of the complete trajectory. The solution for each of these segments was used to generate a biased population that would serve as the initial population to solve the next problem, corresponding to the next segment. For this experiment, it is proposed to establish a linear trajectory that is a common approximation in different fields, which is denoted by (15):

$$P_i = \begin{pmatrix} x_{d,i} \\ y_{d,i} \\ z_{d,i} \end{pmatrix} = P_0 + i\frac{P_k - P_0}{k} \tag{15}$$

In this particular case, the task was divided into $k = 10$ optimization problems, that is, the same number of the runs of the first experiment to verify if it was possible to achieve a better solution with a lower total number of generations in one run with 10 simpler problems. The $k$ optimization problems are described with (16), where $x, y, z$ are calculated by the model generated from the homogeneous transformations.

$$\min f_i(\vec{\theta}) = \sqrt{(x(\vec{\theta}) - x_{d,i})^2 + (y(\vec{\theta}) - y_{d,i})^2 + (z(\vec{\theta}) - z_{d,i})^2}, \qquad i = 1, 2, \ldots, k \tag{16}$$



The 10 individual results for the total solution in the test run are shown in Table 3, including the number of generations required for the calculation, whereas the corresponding graph is presented in Figure 4. Because of the size order of the values for the objective function, its values are presented in scientific notation.

**Table 3.** Data of the test run by the second approximation.

| Problem | Generations | Objective Function (cm) | Distance to the Previous Pose (rad) |
|---|---|---|---|
| 1 | 7844 | $2.9576 \times 10^{-4}$ | 0.1809 |
| 2 | 14,804 | $8.7079 \times 10^{-4}$ | 0.2170 |
| 3 | 14,615 | $9.1547 \times 10^{-4}$ | 0.3579 |
| 4 | 12,387 | $8.4897 \times 10^{-4}$ | 0.1785 |
| 5 | 13,913 | $5.5925 \times 10^{-4}$ | 0.1815 |
| 6 | 25,682 | $6.9012 \times 10^{-4}$ | 0.2210 |
| 7 | 2800 | $6.2471 \times 10^{-4}$ | 0.1649 |
| 8 | 8277 | $6.2898 \times 10^{-4}$ | 0.2034 |
| 9 | 8223 | $8.9564 \times 10^{-4}$ | 0.1157 |
| 10 | 8449 | $6.7832 \times 10^{-4}$ | 0.1062 |

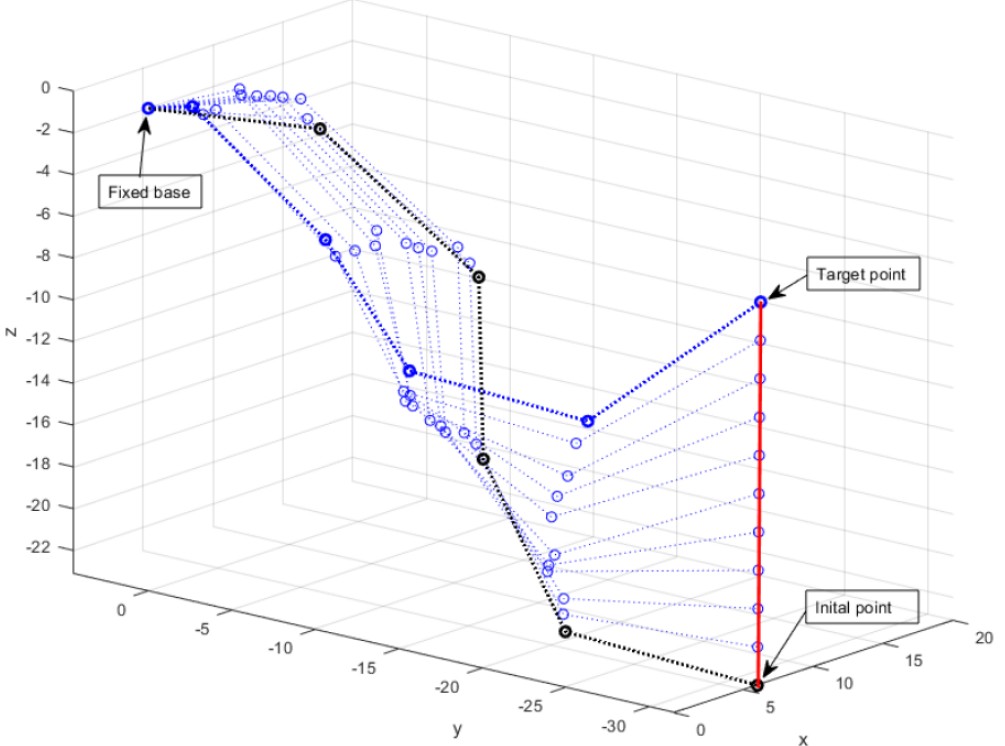

**Figure 4.** Kinematic chain when adopting the individual results obtained for the total solution in the test run, for the second experiment.

The total solution from the test run met the error criterion, and its distance from the original position was 1.0644, which was lower than any of the solutions obtained with the approach in the first experiment. The approach used in the second experiment produced a better result both in reducing the error when reaching the target point and in maintaining a similarity with the initial pose. In addition, the sum of generations or cycles that takes the algorithm to solve the IKP by this approach is lower than the total required in the first experiment. Figure 5 shows the best solutions generated in both experiments.

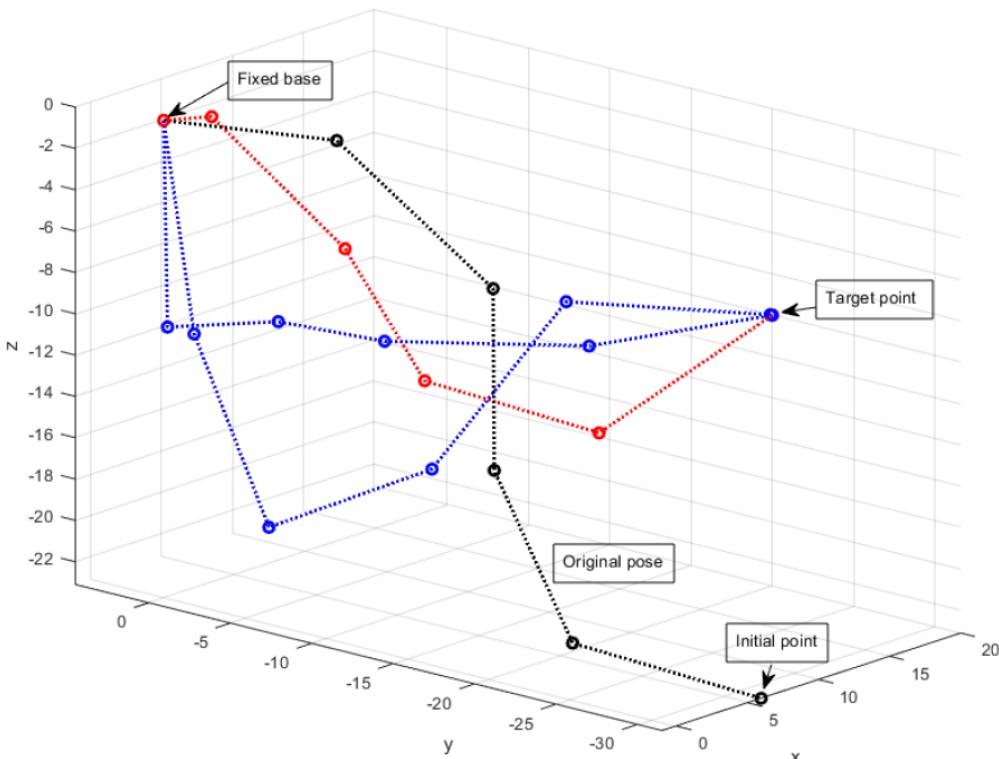

**Figure 5.** Kinematic chain when adopting the best values obtained in Experiment 1 is shown in blue, in red the one obtained by the second approach, and in black the original pose.

Due to the stochastic nature of the heuristic techniques, there is a possibility that the solution shown in the test run with the second experiment represents an extraordinary or difficult case to replicate. Considering this, the experiment was replicated 10 times in order to compare the solutions found in new executions with different random values. The results are shown in Table 4 including the mean and the standard deviation. Again, because of the size order of the values for the objective function, they are presented in scientific notation. Figure 6 shows the total solutions found.

**Table 4.** Results of the second experiment.

| Run | Generations | Objective Function (cm) | Distance from the Original Pose (rad) |
|---|---|---|---|
| 1 | 85,743 | $9.3063 \times 10^{-4}$ | 1.1238 |
| **2** | **119,189** | $\mathbf{9.6545 \times 10^{-4}}$ | **1.1018** |
| 3 | 72,895 | $9.9947 \times 10^{-4}$ | 1.1196 |
| **4** | **82,429** | $\mathbf{6.6060 \times 10^{-4}}$ | **1.3207** |
| 5 | 69,345 | $7.6651 \times 10^{-4}$ | 1.1267 |
| 6 | 82,149 | $7.3738 \times 10^{-4}$ | 1.2138 |
| 7 | 79,054 | $6.6518 \times 10^{-4}$ | 1.1704 |
| 8 | 86,091 | $6.8845 \times 10^{-4}$ | 1.2011 |
| 9 | 89,843 | $8.8097 \times 10^{-4}$ | 1.2603 |
| 10 | 78,608 | $7.4327 \times 10^{-4}$ | 1.1898 |
| Mean | 84,535 | $8.0379 \times 10^{-4}$ | 1.1828 |
| Standard deviation | 13,640 | $1.2865 \times 10^{-4}$ | 0.0697 |

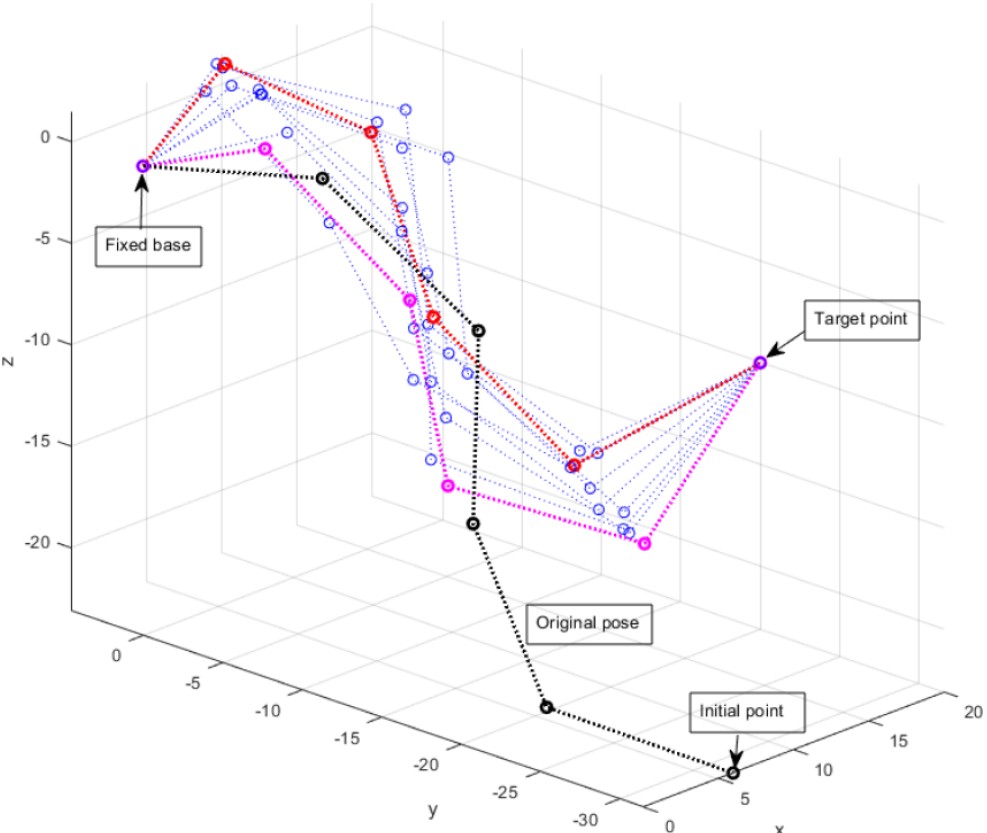

**Figure 6.** Final postures of the kinematic chain in ten executions by the proposed approach; the pose with the greatest similarity shown in red, and the pose with the lowest error in magenta.

Because of the aforementioned nature of the metaheuristic algorithms, there are differences in the number of cycles that were required in each of the runs to solve the IKP. This can be attributed to the random components of the algorithm, but it is clear that all the solutions met the error condition and had higher similarity with the initial pose than that of any of the obtained solutions in the first experiment. In addition, Figure 6 shows that the solutions from the second experiment had a higher similarity among themselves, contrasted with those obtained in the first experiment (Figure 3).

### 3.2. Obtaining the Sequence of Configurations along a Circular Trajectory

Considering the quality of the previous results, the proposed approach could be applied to the solution of the IKP involving trajectories with higher complexity than the one used in the second experiment (a straight line). For this reason, in the third experiment, a circular trajectory with 50 points was proposed. The results are shown in Figure 7.

In this experiment, a solution that met the error criteria was found for each problem associated the algorithm. A similar position was also maintained throughout the trajectory due to the manipulation carried out on the initial population of each problem. These results show that the methodology applied tries to maintain a similarity with the previous problem, but given a succession of problems that lead to the initial point as a consequence of traversing a trajectory, it does not imply that the position was maintained, only that it preserved some degree of similarity.

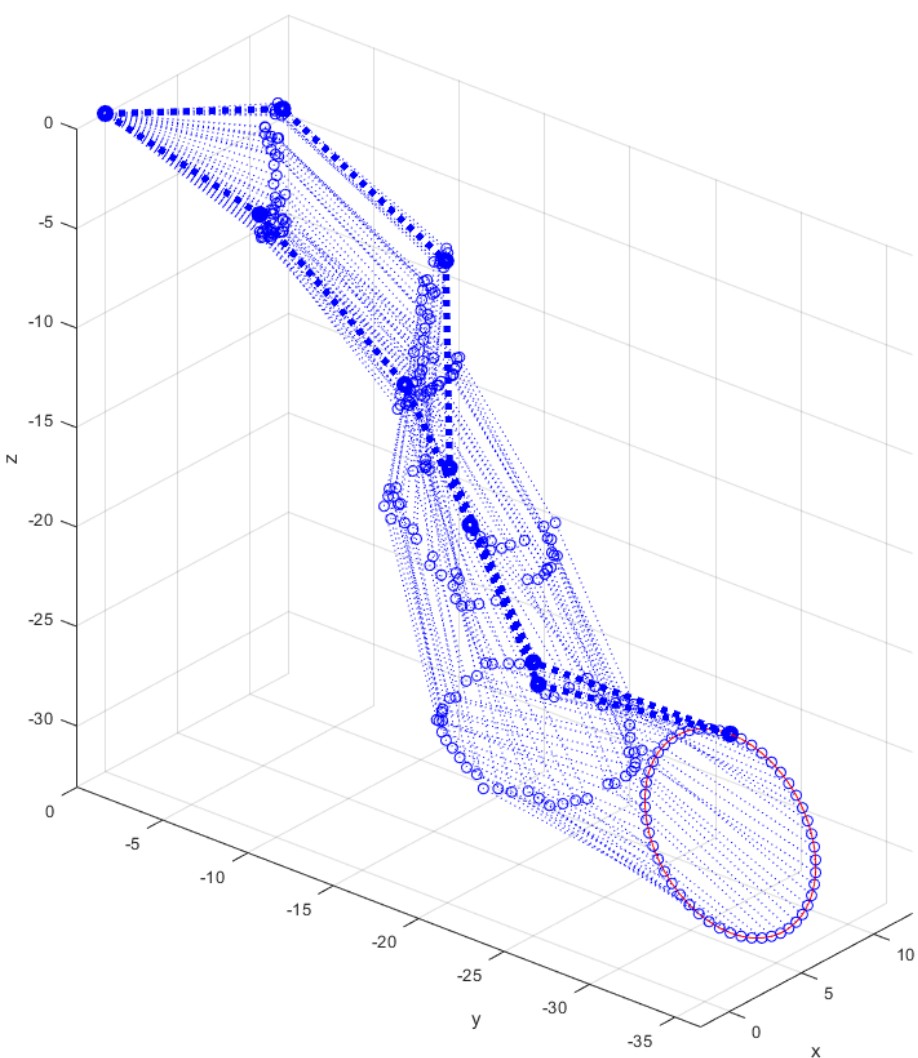

**Figure 7.** Solutions for the optimization problems of the circular trajectory. The thick lines correspond to the initial and final postures.

As shown in Figure 7, the final and initial postures were not the same. This could be an undesired condition for cyclic tasks, since for this experiment the start and end of a sequence of configurations are required to have the same value, also for every pair of subsequent configurations must be similar. Two approaches are proposed to achieve this, modifying the strategy to bias the population. The first one consists in biasing the population around the initial configuration (values corresponding to the initial pose) for all the optimization problems, as it is shown in (17), where $pbest_0$ corresponds to the the initial pose. The results of this approximation are shown in Figure 8, where the initial and final poses presented higher similarity in comparison with the results in Figure 7.

$$pop_k = pbest_0 + M(rand - 0.5) \tag{17}$$

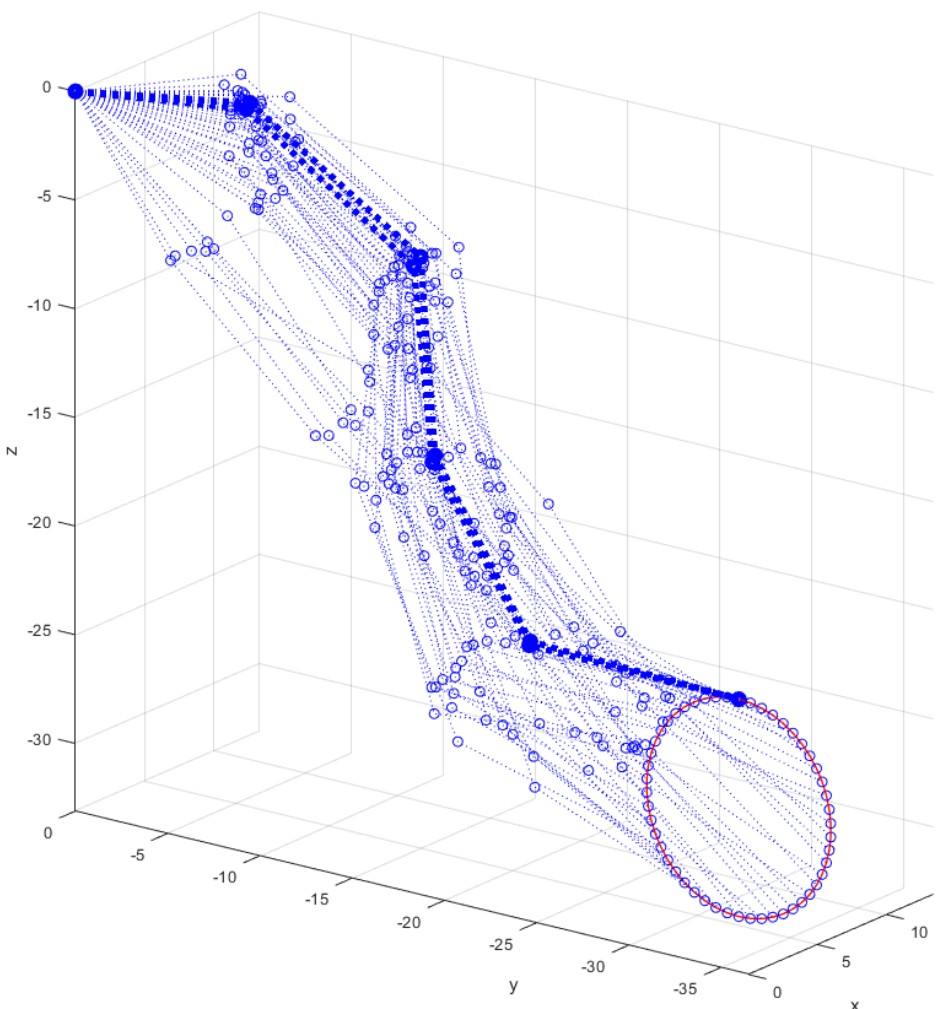

**Figure 8.** Solutions for the optimization problems of the circular trajectory biasing the population around the values of the initial pose. The thick lines correspond to the initial and final posture.

The previous approach achieved the desired result, in the sense that the similarity between the initial and final poses was increased, but this proposal lost the information obtained from the solution of the previous problem, thus increasing the probability that two successive configurations would present low similarity. It is proposed to average the initial position with the best individual of the previous problem to balance this aspect, as it is shown in (18).

$$pop_k = 0.5(pbest_0 + best_{k-1}) + M(rand - 0.5) \tag{18}$$

The results of this approximation are shown in Figure 9. In this case, not only did the initial and final poses present a higher similarity compared to the result in Figure 7, but also every pair in the set of solutions behaved in the same way.

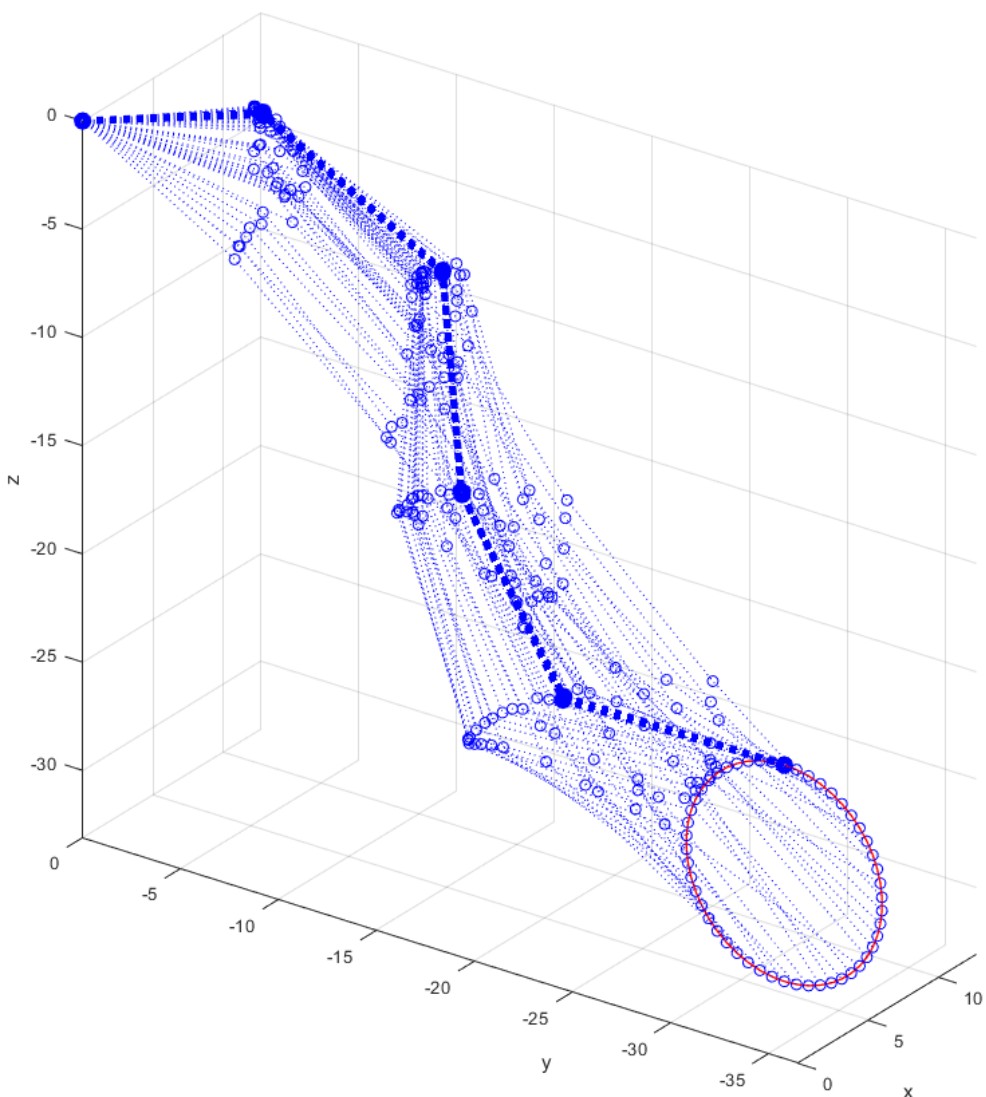

**Figure 9.** Solutions for the optimization problems of the circular trajectory biasing the population around the mean of the values from previous best solution and the initial pose. The thick lines correspond to the initial and final posture.

### 3.3. Obtaining the Sequence of Configurations along a Random Trajectory

Lastly, in the fourth experiment, a random trajectory with 50 points was generated by performing random displacements from the initial position. As in the second and third experiments, for each associated problem, the algorithm found a solution that met the error criteria, and a similar position was maintained throughout the trajectory. The results are shown in Figure 10.

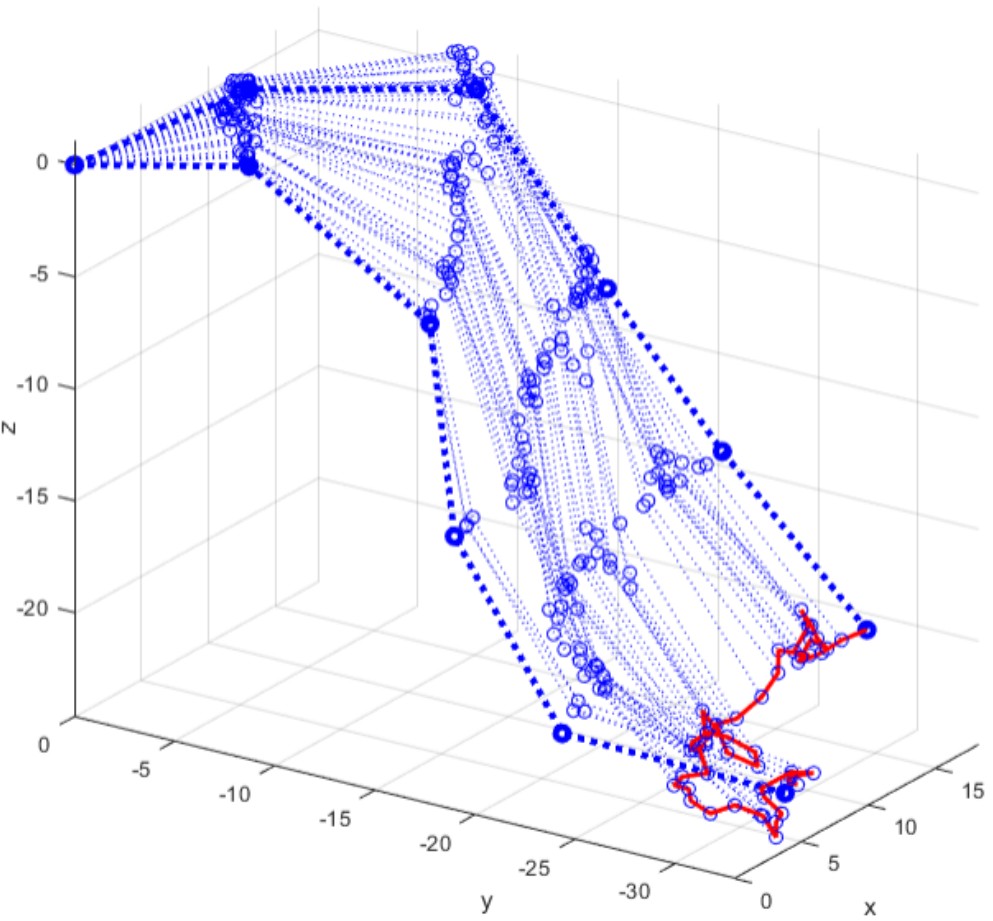

**Figure 10.** Solutions for the optimization problems of the random path. The thick lines correspond to the initial and final posture.

## 4. Conclusions

In this work, a new approach to solve the IKP was proposed that consists in solving it as a sequence of single optimization problems using a modified DE, with each problem associated with a small segment of a discretized path that connects the initial position of the kinematic chain with the desired position. As another contribution of this proposal, the single problems were solved, biasing their initial populations to produce a specific behavior where a single solution has a high similarity with the solution found in the previous problem. Two additional modifications were included to increase the similarity when solving the IKP for close trajectories. This implies that the poses adopted by the kinematic chain along the trajectory (corresponding to the total solution) had a high similarity with the initial pose. The proposed approach was tested by solving the IKP for a 5-link 10-DOF kinematic chain applying the differential evolution algorithm, and the results were compared with the solutions obtained by a traditional approximation without dividing the trajectory in different experiments considering a standard error condition.

The results generated by the proposal when solving the IKP fulfilled the error condition. Particularly, the proposed approach required fewer execution cycles to obtain better solutions compared to the traditional approach with an undivided trajectory also solved with DE. In addition, it is useful as a method for trajectory tracking in which the succession of similar configurations can help in establishing a more complete trajectory without abrupt changes in the angular values of the joints. One of the advantages of this approach is that it does not have to deal with the issue of singularities that affect the behavior of some analytic methods, such as division by zero or multiple solutions that produce an undecidable problem, since each position explored by the algorithm is a mapping of the angular values to a position in $\mathbb{R}^3$.

The proposed approach was applied to solve the IKP for a pure serial kinematic chain, that is, without considering physical aspects and their implications (e.g., singularities). Therefore, as future work, it is proposed to take those aspects into account in the research, for example, by incorporating them as constraints in the optimization problem and including a constraint handler in DE, such as the feasibility rules [23]. However, the method may not be applicable to cases such as certain types of parallel robots since, in some configurations, there is no function to obtain the position and orientation coordinates of the end effector from the angles of the actuator's FK. For those cases, a different approach has been developed, where the IKP is solved taking as a base the FK, which is, in turn, calculated with diverse tools such as mathematical programming or metaheuristics. Other considerations may fall into analyzing the problem as a multiobjective optimization case, where other objective functions are sought to minimize the movement of the center of gravity or the displacement of the effectors prioritizing rotations of the base with less movement to consider energy costs. For example, two objectives can be considered: positioning the end effector in a desired position (minimize the mapped value by the FK) and maximizing the similarity with the start configuration. As part of the analysis, it is required to determine if the objective functions are opposite to each other.

On the other hand, considering that an important part of the proposal presented in this work consisted in biasing the initial population that DE used to solve the sequence of optimization problems, other metaheuristics can be applied to evaluate its performance with biased populations and other modifications to maintain and/or improve the results obtained. Another metaheuristics can be applied if constraints are included in the modeling of the optimization problems, since DE has difficulties in finding feasible solutions with equality constraints.

**Author Contributions:** Conceptualization, R.L.-M., E.A.P.-F. and L.G.C.-R.; methodology, R.L.-M., E.A.P.-F. and L.G.C.-R.; software, M.C.M.-R.; validation, E.V.-A. and M.C.M.-R.; formal analysis, E.V.-A. and M.C.M.-R.; investigation, R.L.-M.; writing—original draft preparation, R.L.-M. and E.V.-A.; writing—review and editing, R.L.-M. and E.V.-A. All authors have read and agreed to the published version of the manuscript.

**Funding:** This research received no external funding.

**Institutional Review Board Statement:** Not applicable.

**Informed Consent Statement:** Not applicable.

**Data Availability Statement:** The data and results generated in this project are available in https://github.com/ra0514/IKP-MDE.

**Acknowledgments:** The authors wish to thank the Instituto Politécnico Nacional of México, for its support both via Secretaría de Investigación y Posgrado with the SIP projects. Raúl López-Muñoz would like to thank CONACYT of México for their doctorate grant.

**Conflicts of Interest:** The authors declare no conflict of interest.

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
