# Peer review of "Inverse Kinematics: An Alternative Solution Approach Applying Metaheuristics"

_applsci, doi:10.3390/app13116543_

Round 1

Reviewer 1 Report

Interesting article. I don't really have any major criticisms other than a few nature of the data presentation. Figure 1 is made unsightly. In general, it is unnecessary, everyone knows the rules for fixing coordinate systems according to the Denavit-Hartenberg notation. There is no consistency in the way of presenting numerical values in tables (format, number of significant digits).

Reviewer 2 Report

The article has presented an alternative solution approach to solve the inverse kinematics problem of robotic manipulators using metaheuristics. This manuscript could be accepted for publication after considering the below-mentioned points:

·         Please highlight the novelty, major findings and conclusions. Three main points could be included clearly in the text, especially in abstract and conclusions sections:

1.       novelty, 2. contribution and importance, 3. Applicability

·         Please change Figure 1 to an original high-quality figure.

·         In Section 2 Please provide a step-by-step flowchart for optimization approach including selection of the metaheuristic technique and modeling considerations, and the experiments to final result.

·         Please describe how Equations 15-17 were used in your step-by-step flowchart.

The flowchart is necessarily needed to show the repeatability of the method and shows that your research is applicable to other mechanisms, especially parallel ones that have more limitations compared to the serials.

·         In complex parallel mechanisms like Hexarot and Hexapod, the joint constraints and singularities should be considered in the kinematic models, as they make limitations for the links to move within the workspace of the mechanism. This has been stated in the conclusion section* as a future work, however, to improve the quality of paper it would be better to discuss more and clear details to show how you wish to consider joint constraints and singularities. This, moreover, adds to the contribution, importance, and applicability of your method.

*The proposed approach was applied to solve the IKP for pure kinematic chains, this is, without considering physical aspects and their implications. These considerations may fall into modeling the problem as a multiobjetive optimization case, where other objective functions are sought to minimize the movement of the center of gravity or the displacement of the effectors prioritizing rotations of the base with less movement to consider energy costs.

Round 2

Reviewer 2 Report

The authors well-responded the comments, and the manuscript can be accepted in the present form.